# Professor Forcing: A New Algorithm for Training Recurrent Networks

**Anirudh Goyal,*** **Alex Lamb***, **Ying Zhang, Saizheng Zhang,**
**Aaron Courville and Yoshua Bengio**[1]
MILA, Université de Montréal, [1]CIFAR
{anirudhgoyal9119, alex6200, ying.zhlisa, saizhenglisa,
aaron.courville, yoshua.umontreal}@gmail.com

## Abstract

The Teacher Forcing algorithm trains recurrent networks by supplying observed sequence values as inputs during training and using the network's own one-step-ahead predictions to do multi-step sampling. We introduce the Professor Forcing algorithm, which uses adversarial domain adaptation to encourage the dynamics of the recurrent network to be the same when training the network and when sampling from the network over multiple time steps. We apply Professor Forcing to language modeling, vocal synthesis on raw waveforms, handwriting generation, and image generation. Empirically we find that Professor Forcing acts as a regularizer, improving test likelihood on character level Penn Treebank and sequential MNIST. We also find that the model qualitatively improves samples, especially when sampling for a large number of time steps. This is supported by human evaluation of sample quality. Trade-offs between Professor Forcing and Scheduled Sampling are discussed. We produce T-SNEs showing that Professor Forcing successfully makes the dynamics of the network during training and sampling more similar.

## 1  Introduction

Recurrent neural networks (RNNs) have become to be the generative models of choice for sequential data (Graves, 2012) with impressive results in language modeling (Mikolov, 2010; Mikolov and Zweig, 2012), speech recognition (Bahdanau *et al.*, 2015; Chorowski *et al.*, 2015), Machine Translation (Cho *et al.*, 2014a; Sutskever *et al.*, 2014; Bahdanau *et al.*, 2014), handwriting generation (Graves, 2013), image caption generation (Xu *et al.*, 2015; Chen and Lawrence Zitnick, 2015), etc.

The RNN models the data via a fully-observed directed graphical model: it decomposes the distribution over the discrete time sequence $y_1, y_2, \ldots y_T$ into an ordered product of conditional distributions over tokens

$$P(y_1, y_2, \ldots y_T) = P(y_1) \prod_{t=1}^{T} P(y_t \mid y_1, \ldots y_{t-1}).$$

By far the most popular training strategy is via the maximum likelihood principle. In the RNN literature, this form of training is also known as *teacher forcing* (Williams and Zipser, 1989), due to the use of the ground-truth samples $y_t$ being fed back into the model to be conditioned on for the prediction of later outputs. These fed back samples force the RNN to stay close to the ground-truth sequence.

When using the RNN for prediction, the ground-truth sequence is not available conditioning and we sample from the joint distribution over the sequence by sampling each $y_t$ from its conditional

distribution given the previously generated samples. Unfortunately, this procedure can result in problems in generation as small prediction error compound in the conditioning context. This can lead to poor prediction performance as the RNN's conditioning context (the sequence of previously generated samples) diverge from sequences seen during training.

Recently, (Bengio *et al.*, 2015) proposed to remedy that issue by mixing two kinds of inputs during training: those from the ground-truth training sequence and those generated from the model. However, when the model generates several consecutive $y_t$'s, it is not clear anymore that the correct target (in terms of its distribution) remains the one in the ground truth sequence. This is mitigated in various ways, by making the self-generated subsequences short and annealing the probability of using self-generated vs ground truth samples. However, as remarked by Huszár (2015), scheduled sampling yields a biased estimator, in that even as the number of examples and the capacity go to infinity, this procedure may not converge to the correct model. It is however good to note that experiments with scheduled sampling clearly showed some improvements in terms of the robustness of the generated sequences, suggesting that something indeed needs to be fixed (or replaced) with maximum-likelihood (or teacher forcing) training of generative RNNs.

In this paper, we propose an alternative way of training RNNs which explicitly seeks to make the generative behavior and the teacher-forced behavior match as closely as possible. This is particularly important to allow the RNN to continue generating robustly well beyond the length of the sequences it saw during training. More generally, we argue that this approach helps to better model long-term dependencies by using a training objective that is not solely focused on predicting the next observation, one step at a time.

Our work provides the following contributions regarding this new training framework:

- We introduce a novel method for training generative RNNs called Professor Forcing, meant to improve long-term sequence sampling from recurrent networks. We demonstrate this with human evaluation of sample quality by performing a study with human evaluators.

- We find that Professor Forcing can act as a regularizer for recurrent networks. This is demonstrated by achieving improvements in test likelihood on character-level Penn Treebank, Sequential MNIST Generation, and speech synthesis. Interestingly, we also find that training performance can also be improved, and we conjecture that it is because longer-term dependencies can be more easily captured.

- When running an RNN in sampling mode, the region occupied by the hidden states of the network diverges from the region occupied when doing teacher forcing. We empirically study this phenomenon using T-SNEs and show that it can be mitigated by using Professor Forcing.

- In some domains the sequences available at training time are shorter than the sequences that we want to generate at test time. This is usually the case in long-term forecasting tasks (climate modeling, econometrics). We show how using Professor Forcing can be used to improve performance in this setting. Note that scheduled sampling cannot be used for this task, because it still uses the observed sequence as targets for the network.

## 2 Proposed Approach: Professor Forcing

The basic idea of Professor Forcing is simple: while we do want the generative RNN to match the training data, we also want the behavior of the network (both in its outputs and in the dynamics of its hidden states) to be indistinguishable whether the network is trained with its inputs clamped to a training sequence (teacher forcing mode) or whether its inputs are self-generated (free-running generative mode). Because we can only compare the distribution of these sequences, it makes sense to take advantage of the generative adversarial networks (GANs) framework (Goodfellow *et al.*, 2014) to achieve that second objective of matching the two distributions over sequences (the one observed in teacher forcing mode vs the one observed in free-running mode).

Hence, in addition to the generative RNN, we will train a second model, which we call the discriminator, and that can also process variable length inputs. In the experiments we use a bidirectional RNN architecture for the discriminator, so that it can combine evidence at each time step $t$ from the past of the behavior sequence as well as from the future of that sequence.

## 2.1 Definitions and Notation

Let the training distribution provide $(\boldsymbol{x}, \boldsymbol{y})$ pairs of input and output sequences (possibly there are no inputs at all). An output sequence $\boldsymbol{y}$ can also be generated by the generator RNN when given an input sequence $\boldsymbol{x}$, according to the sequence to sequence model distribution $P_{\boldsymbol{\theta}_g}(\boldsymbol{y}|\boldsymbol{x})$. Let $\boldsymbol{\theta}_g$ be the parameters of the generative RNN and $\boldsymbol{\theta}_d$ be the parameters of the discriminator. The discriminator is trained as a probabilistic classifier that takes as input a behavior sequence $\boldsymbol{b}$ derived from the generative RNN's activity (hiddens and outputs) when it either generates or is constrained by a sequence $\boldsymbol{y}$, possibly in the context of an input sequence $\boldsymbol{x}$ (often but not necessarily of the same length). The behavior sequence $\boldsymbol{b}$ is either the result of running the generative RNN in teacher forcing mode (with $\boldsymbol{y}$ from a training sequence with input $\boldsymbol{x}$), or in free-running mode (with $\boldsymbol{y}$ self-generated according to $P_{\boldsymbol{\theta}_g}(\boldsymbol{y}|\boldsymbol{x})$, with $\boldsymbol{x}$ from the training sequence). The function $B(\boldsymbol{x}, \boldsymbol{y}, \boldsymbol{\theta}_g)$ outputs the behavior sequence (chosen hidden states and output values) given the appropriate data (where $\boldsymbol{x}$ always comes from the training data but $\boldsymbol{y}$ either comes from the data or is self-generated). Let $D(\boldsymbol{b})$ be the output of the discriminator, estimating the probability that $\boldsymbol{b}$ was produced in teacher-forcing mode, given that half of the examples seen by the discriminator are generated in teacher forcing mode and half are generated in the free-running mode.

Note that in the case where the generator RNN does not have any conditioning input, the sequence $\boldsymbol{x}$ is empty. Note also that the generated output sequences could have a different length then the conditioning sequence, depending of the task at hand.

## 2.2 Training Objective

The discriminator parameters $\boldsymbol{\theta}_d$ are trained as one would expect, i.e., to maximize the likelihood of correctly classifying a behavior sequence:

$$C_d(\boldsymbol{\theta}_d|\boldsymbol{\theta}_g) = E_{(\boldsymbol{x},\boldsymbol{y})\sim\text{data}}[-\log D(B(\boldsymbol{x},\boldsymbol{y},\boldsymbol{\theta}_g),\boldsymbol{\theta}_d) + E_{\boldsymbol{y}\sim P_{\boldsymbol{\theta}_g}(\boldsymbol{y}|\boldsymbol{x})}[-\log(1-D(B(\boldsymbol{x},\boldsymbol{y},\boldsymbol{\theta}_g),\boldsymbol{\theta}_d)]]. \tag{1}$$

Practically, this is achieved with a variant of stochastic gradient descent with minibatches formed by combining $N$ sequences obtained in teacher-forcing mode and $N$ sequences obtained in free-running mode, with $\boldsymbol{y}$ sampled from $P_{\boldsymbol{\theta}_g}(\boldsymbol{y}|\boldsymbol{x})$. Note also that as $\boldsymbol{\theta}_g$ changes, the task optimized by the discriminator changes too, and it has to track the generator, as in other GAN setups, hence the notation $C_d(\boldsymbol{\theta}_d|\boldsymbol{\theta}_g)$.

The generator RNN parameters $\boldsymbol{\theta}_g$ are trained to (a) maximize the likelihood of the data and (b) fool the discriminator. We considered two variants of the latter. The negative log-likelihood objective (a) is the usual teacher-forced training criterion for RNNs:

$$NLL(\boldsymbol{\theta}_g) = E_{(\boldsymbol{x},\boldsymbol{y})\sim\text{data}}[-\log P_{\boldsymbol{\theta}_g}(\boldsymbol{y}|\boldsymbol{x})]. \tag{2}$$

Regarding (b) we consider a training objective that only tries to change the free-running behavior so that it better matches the teacher-forced behavior, considering the latter fixed:

$$C_f(\boldsymbol{\theta}_g|\boldsymbol{\theta}_d) = E_{\boldsymbol{x}\sim\text{data},\boldsymbol{y}\sim P_{\boldsymbol{\theta}_g}(\boldsymbol{y}|\boldsymbol{x})}[-\log D(B(\boldsymbol{x},\boldsymbol{y},\boldsymbol{\theta}_g),\boldsymbol{\theta}_d)]. \tag{3}$$

In addition (and optionally), we can ask the teacher-forced behavior to be indistinguishable from the free-running behavior:

$$C_t(\boldsymbol{\theta}_g|\boldsymbol{\theta}_d) = E_{(\boldsymbol{x},\boldsymbol{y})\sim\text{data}}[-\log(1 - D(B(\boldsymbol{x},\boldsymbol{y},\boldsymbol{\theta}_g),\boldsymbol{\theta}_d))]. \tag{4}$$

In our experiments we either perform stochastic gradient steps on $NLL + C_f$ or on $NLL + C_f + C_t$ to update the generative RNN parameters, while we always do gradient steps on $C_d$ to update the discriminator parameters.

## 3 Related Work

Professor Forcing is an adversarial method for learning generative models that is closely related to Generative Adversarial Networks (Goodfellow *et al.*, 2014) and Adversarial Domain Adaptation Ajakan *et al.* (2014); Ganin *et al.* (2015). Our approach is similar to generative adversarial networks (GANs) because both use a discriminative classifier to provide gradients for training a generative model. However, Professor Forcing is different because the classifier discriminates between hidden

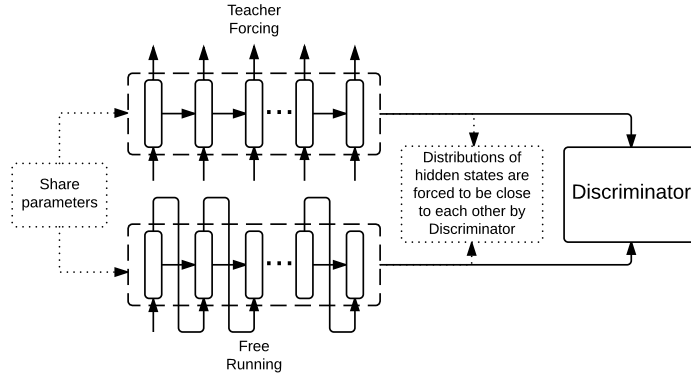

Figure 1: Architecture of the Professor Forcing - Learn correct one-step predictions such as to to obtain the same kind of recurrent neural network dynamics whether in open loop (teacher forcing) mode or in closed loop (generative) mode. An open loop generator that does one-step-ahead prediction correctly. Recursively composing these outputs does multi-step prediction (closed-loop) and can generate new sequences. This is achieved by train a classifier to distinguish open loop (teacher forcing) vs. closed loop (free running) dynamics, as a function of the sequence of hidden states and outputs. Optimize the closed loop generator to fool the classifier. Optimize the open loop generator with teacher forcing. The closed loop and open loop generators share all parameters

states from sampling mode and teacher forcing mode, whereas the GAN's classifier discriminates between real samples and generated samples. One practical advantage of Professor Forcing over GANs is that Professor Forcing can be used to learn a generative model over discrete random variables without requiring to approximate backpropagation through discrete spaces Bengio *et al.* (2013).

The Adversarial Domain Adaptation uses a classifier to discriminate between the hidden states of the network with inputs from the source domain and the hidden states of the network with inputs from the target domain. However this method was not applied in the context of generative models, more specifically, was not applied to the task of improving long-term generation from recurrent networks.

Alternative non-adversarial methods have been explored for improving long-term generation from recurrent networks. The scheduled sampling method Bengio *et al.* (2015), which is closely related to SEARN (Daumé *et al.*, 2009) and DAGGER Ross *et al.* (2010), involves randomly using the network's predictions as its inputs (as in sampling mode) with some probability that increases over the course of training. This forces the network to be able to stay in a reasonable regime when receiving the network's predictions as inputs instead of observed inputs. While Scheduled Sampling shows improvement on some tasks, it is not a consistent estimation strategy. This limitation arises because the outputs sampled from the network could correspond to a distribution that is not consistent with the sequence that the network is trained to generate. This issue is discussed in detail in Huszár (2015). A practical advantage of Scheduled Sampling over Professor Forcing is that Scheduled Sampling does not require the additional overhead of having to train a discriminator network.

Finally, the idea of matching the behavior of the model when it is generating in a free-running way with its behavior when it is constrained by the observed data (being clamped on the "visible units") is precisely that which one obtains when zeroing the maximum likelihood gradient on undirected graphical models with latent variables such as the Boltzmann machine. Training Boltzmann machines amounts to matching the sufficient statistics (which summarize the behavior of the model) in both "teacher forced" (positive phase) and "free-running" (negative phase) modes.

## 4 Experiments

### 4.1 Networks Architecture and Professor Forcing Setup

The neural networks and Professor Forcing setup used in the experiments is the following. The generative RNN has single hidden layer of gated recurrent units (GRU), previously introduced by (Cho *et al.*, 2014b) as a computationally cheaper alternative to LSTM units (Hochreiter and Schmidhuber, 1997). At each time step, the generative RNN reads an element $x_t$ of the input

sequence (if any) and an element of the output sequence $y_t$ (which either comes from the training data or was generated at the previous step by the RNN). It then updates its state $h_t$ as a function of its previous state $h_{t-1}$ and of the current input $(x_t, y_t)$. It then computes a probability distribution $P_{\theta_g}(y_{t+1}|h_t) = P_{\theta_g}(y_{t+1}|x_1, \ldots, x_t, y_1, \ldots, y_t)$ over the next element of the output. For discrete outputs this is achieved by a softmax / affine layer on top of $h_t$, with as many outputs as the size of the set of values that $y_t$ can take. In free-running mode, $y_{t+1}$ is then sampled from this distribution and will be used as part of the input for the next time step. Otherwise, the ground truth $y_t$ is used.

The behavior function $B$ used in the experiments outputs the pre-tanh activation of the GRU states for the whole sequence considered, and optionally the softmax outputs for the next-step prediction, again for the whole sequence.

The discriminator architecture we used for these experiments is based on a bidirectional recurrent neural network, which comprises two RNNs (again, two GRU networks), one running forward in time on top of the input sequence $b$, and one running backwards in time, with the same input. The hidden states of these two RNNs are concatenated at each time step and fed to a multi-layer neural network shared across time (the same network is used for all time steps). That MLP has three layers, each composing an affine transformation and a rectifier (ReLU). Finally, the output layer composes an affine transformation and a sigmoid that outputs $D(b)$.

When the discriminator is too poor, the gradient it propagates into the generator RNN could be detrimental. For this reason, we back-propagate from the discriminator into the generator RNN only when the discriminator classification accuracy is greater than 75%. On the other hand, when the discriminator is too successful at identifying fake inputs, we found that it would also hurt to continue training it. So when its accuracy is greater than 99%, we do not update the discriminator.

Both networks are trained by minibatch stochastic gradient descent with adaptive learning rates and momentum determined by the Adam algorithm (Kingma and Ba, 2014). All of our experiments were implemented using the Theano framework (Al-Rfou *et al.*, 2016).

## 4.2 Character-Level Language Modeling

We evaluate Professor Forcing on character-level language modeling on Penn-Treebank corpus, which has an alphabet size of 50 and consists of 5059k characters for training, 396k characters for validation and 446k characters for test. We divide the training set into non-overlapping sequences with each length of 500. During training, we monitor the negative log-likelihood (NLL) of the output sequences. The final model are evaluated by bits-per-character (BPC) metric. The generative RNN

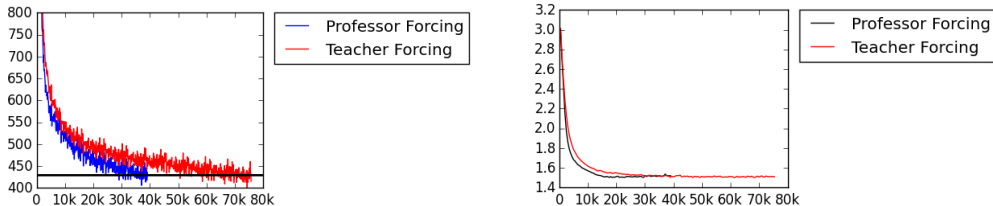

Figure 2: Penn Treebank Likelihood Curves in terms of the number of iterations. Training Negative Log-Likelihood (left). Validation BPC (Right)

implements an 1 hidden layer GRU with 1024 hidden units. We use Adam algorithm for optimization with a learning rate of 0.0001. We feed both the hidden states and char level embeddings into the discriminator. All the layers in the discriminator consists of 2048 hidden units. Output activation of the last layer is clipped between -10 and 10. We see that training cost of Professor Forcing network decreases faster compared to teacher forcing network. The training time of our model is 3 times more as compared to teacher forcing, since our model includes sampling phase, as well as passing the hidden distributions corresponding to free running and teacher forcing phase to the discriminator. The final BPC on validation set using our baseline was 1.50 while using professor forcing it is 1.48.

On word level Penn Treebank we did not observe any difference between Teacher Forcing and Professor Forcing. One possible explanation for this difference is the increased importance of long-term dependencies in character-level language modeling.

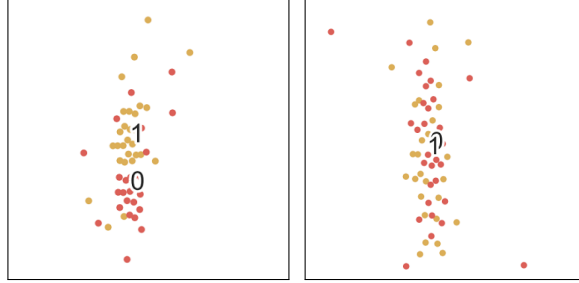

Figure 3: T-SNE visualization of hidden states, left: with teacher forcing, right: with professor forcing. Red dots correspond to teacher forcing hidden states, while the gold dots correspond to free running mode. At t = 500, the closed-loop and open-loop hidden states clearly occupy distinct regions with teacher forcing, meaning that the network enters a region during sampling distinct from the region seen during teacher forcing training. With professor forcing, these regions now largely overlap. We computed 30 T-SNEs for Teacher Forcing and 30 T-SNEs for Professor Forcing and found that the mean centroid distance was reduced from 3000 to 1800 (40% relative reduction). The mean distance from a hidden state in the training network to a hidden state in the sampling network was reduced from 22.8 with Teacher Forcing to 16.4 with Professor Forcing (vocal synthesis).

| Method | MNIST NLL |
|---|---|
| DBN 2hl (Germain *et al.*, 2015) | $\approx 84.55$ |
| NADE (Larochelle and Murray, 2011) | 88.33 |
| EoNADE-5 2hl (Raiko *et al.*, 2014) | 84.68 |
| DLGM 8 leapfrog steps (Salimans *et al.*, 2014) | $\approx 85.51$ |
| DARN 1hl (Gregor *et al.*, 2015) | $\approx 84.13$ |
| DRAW (Gregor *et al.*, 2015) | $\leq 80.97$ |
| Pixel RNN (van den Oord *et al.*, 2016) | **79.2** |
| Professor Forcing (ours) | 79.58 |

Table 1: Test set negative log-likelihood evaluations on Sequential MNIST.

## 4.3 Sequential MNIST

We evaluated Professor Forcing on the task of sequentially generating the pixels in MNIST digits. We use the standard binarized MNIST dataset Murray and Salakhutdinov (2009). We selected hyperparameters for our model on the validation set and elected to use 512 hidden states and a learning rate of 0.0001. For all experiments we used a 3-layer GRU as our generator. Unlike our other experiments, we used a convolutional network for the discriminator instead of a bi-directional RNN, as the pixels have a 2D spatial structure. In Table 1, We note that our model achieves the second best reported likelihood on this task, after the PixelRNN, which used a significantly more complicated architecture for its generator van den Oord *et al.* (2016). Combining Professor Forcing with the PixelRNN would be an interesting area for future research. However, the PixelRNN parallelizes computation in the teacher forcing network in a way that doesn't work in the sampling network. Because Professor Forcing requires running the sampling network during training, naively combining Professor Forcing with the PixelRNN would be very slow.

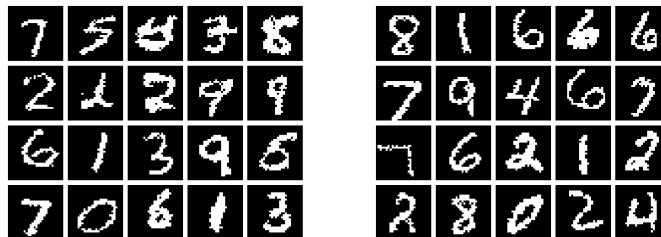

Figure 4: Samples with Teacher Forcing (left) and Professor Forcing (right) on Sequential MNIST.

| Response | Percent | Count |
|---|---|---|
| Professor Forcing Much Better | 19.7 | 151 |
| Professor Forcing Slightly Better | 57.2 | 439 |
| Teacher Forcing Slightly Better | 18.9 | 145 |
| Teacher Forcing Much Better | 4.3 | 33 |
| Total | 100.0 | 768 |

Table 2: Human Evaluation Study Results for Handwriting Generation.

## 4.4 Handwriting Generation

With this task we wanted to investigate if Professor Forcing could be used to perform domain adaptation from a training set with short sequences to sampling much longer sequences. We train the Teacher Forcing model on only 50 steps of text-conditioned handwriting (corresponding to a few letters) and then sample for 1000 time steps . We let the model learn a sequence of (x, y) coordinates together with binary indicators of pen-up vs. pen-down, using the standard handwriting IAM-OnDB dataset, which consists of 13,040 handwritten lines written by 500 writers Liwicki and Bunke (2005). For our teacher forcing model, we use the open source implementation Brebisson (2016) and use their hyperparameters which is based on the model in Graves (2013). For the professor forcing model, we sample for 1000 time steps and run a separate discriminator on non-overlapping segments of length 50 (the number of steps used in the teacher forcing model).

We performed a human evaluation study on handwriting samples. We gave 48 volunteers 16 randomly selected Prof. Forcing samples randomly paired with 16 Teacher Forcing samples and asked them to indicate which sample was higher quality and whether it was "much better" or "slightly better". Both models had equal training time and samples were drawn using the same procedure. Volunteers were not aware of which samples came from which model, see Table 2 for results.

## 4.5 Music Synthesis on Raw Waveforms

We considered the task of vocal synthesis on raw waveforms. For this task we used three hours of monk chanting audio scraped from YouTube (`https://www.youtube.com/watch?v=9-pD28iSiTU`). We sampled the audio at a rate of 1 kHz and took four seconds for each training and validation example. On each time step of the raw audio waveform we binned the signal's value into 8000 bins with boundaries drawn uniformly between the smallest and largest signal values in the dataset. We then model the raw audio waveform as a 4000-length sequence with 8000 potential values on each time step.

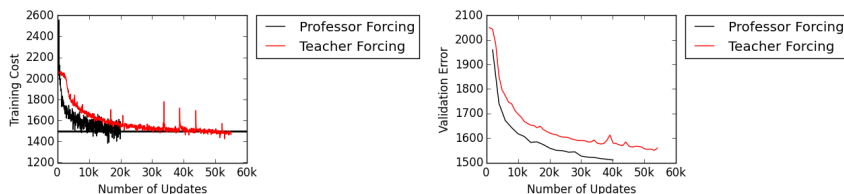

Figure 6: Music Synthesis. Left: training likelihood curves. Right: validation likelihood curves.

We evaluated the quality of our vocal synthesis model using two criteria. First, we demonstrated a regularizing effect and improvement in negative log-likelihood. Second, we observed improvement in the quality of samples. We included a few randomly selected samples in the supplementary material and also performed human evaluation of the samples.

Visual inspection of samples is known to be a flawed method for evaluating generative models, because a generative model could simply memorize a small number of examples from the training set (or slightly modified examples from the training set) and achieve high sample quality. This issue was discussed in Theis *et al.* (2015). However, this is unlikely to be an issue with our evaluation because our method also improved validation set likelihood, whereas a model that achieves quality samples by dropping coverage would have poorer validation set likelihood.

We performed human evaluation by asking 29 volunteers to listen to five randomly selected teacher forcing samples and five randomly selected professor forcing samples (included in supplementary materials and then rate each sample from 1-3 on the basis of quality. The annotators were given the samples in random order and were not told which samples came from which algorithm. The human annotators gave the Professor Forcing samples an average score of 2.20, whereas they gave the Teacher Forcing samples an average score of 1.30.

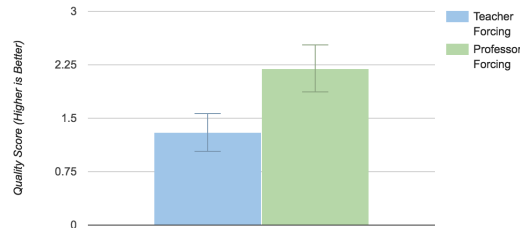

Figure 7: Human evaluator ratings for vocal synthesis samples (higher is better). The height of the bar is the mean of the ratings and the error bar shows the spread of one standard deviation.

## 5    Conclusion

The idea of matching behavior of a model when it is running on its own, making predictions, generating samples, etc. vs when it is forced to be consistent with observed data is an old and powerful one. In this paper we introduce Professor Forcing, a novel instance of this idea when the model of interest is a recurrent generative one, and which relies on training an auxiliary model, the discriminator to spot the differences in behavior between these two modes of behavior. A major motivation for this approach is that the discriminator can look at the statistics of the behavior and not just at the single-step predictions, forcing the generator to behave the same when it is constrained by the data and when it is left generating outputs by itself for sequences that can be much longer than the training sequences. This naturally produces better generalization over sequences that are much longer than the training sequences, as we have found. We have also found that it helped to generalize better in terms of one-step prediction (log-likelihood), even though we are adding a possibly conflicting term to the log-likelihood training objective. This suggests that it acts like a regularizer but a very interesting one because it can also greatly speed up convergence in terms of number of training updates. We validated the advantage of Professor Forcing over traditional teacher forcing on a variety of sequential learning and generative tasks, with particularly impressive results in acoustic generation, where the training sequences are much shorter (because of memory constraints) than the length of the sequences we actually want to generate.

### Acknowledgments

We thank Martin Arjovsky, Dzmitry Bahdanau, Nan Rosemary Ke, José Manuel Rodríguez Sotelo, Alexandre de Brébisson, Olexa Bilaniuk, Hal Daumé III, Kari Torkkola, and David Krueger.

## Footnotes

*Indicates first authors. Ordering determined by coin flip.

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
