[Reviews · NeurIPS 2016]

Reviewer 1

Summary

This work describes a novel algorithm to ensure the dynamics of an LSTM during inference follows that during training. The motivating example is sampling for a long number of steps at test time while only training on shorter sequences at training time. Experimental results are shown on PTB language modelling, MNIST, handwriting generation and music synthesis.

Qualitative Assessment

I found this work to be very readable and easy to understand. Using a discriminator to ensure the dynamics of a LSTM are consistent when training and when sampling is a nice and interesting idea. The experiments seem quite weak on quantitative results. The improvement on language modelling seems very small, however, language does have some special characteristics compared to the other datasets. For one, the output space in language modelling is much greater than the other tasks of handwriting generation, MNIST and music synthesis. Was anything done to try to alleviate this? I like the result on sequential MNIST but it would be good to have quantitative results on handwriting generation and also give final converged numbers for music synthesis test set performance. In addition, the paper repeatedly mentions training on shorter sequences and sampling longer ones but it seems that this was only done for handwriting? It would be interesting to see the results for music synthesis. As described in L257, the adversarial objective seems to add a conflicting term to the objective. Is the adversarial objective of JSD hurting the training with NLL (KL)? Was anything done to explore this? Minor comments: L29: Missing word: 'available conditioning' L32: Missing word: 'error compound' L37: 'not clear anymore that the correct target remains the one in the ground truth sequence...'. What does this mean? The correct target is in the ground truth by definition. L152: Networks -> Network Figure 3: These plots should have borders. What task are the hidden states on? When calculating the mean centroid distance why do this on the T-SNE space? The results would be much stronger on the original space of the hidden states. L243: Extra period. L257: I agree that there is a conflicting term here

Confidence in this Review

2-Confident (read it all; understood it all reasonably well)


Reviewer 2

Summary

The paper is similar to Generative Adversarial Networks (GAN): in addition to a normal sequence model loss function, the parameters try to “fool” a classifier. That classifier is trying to distinguish generated sequences from the sequence model, from real data. A few Objectives are proposed in section 2.2. The key difference to GAN is the B in equations 1-4. B is a function outputs some statistics of the model, such as the hidden state of the RNN, whereas GAN tries rather to discriminate the actual output sequences.

Qualitative Assessment

The method is very close to various other methods. Several of the evaluations only compare with “teacher forcing” (which is just the usual application of the chain rule of probability to build a sequence model, ie. your equation 1). But the other related algorithms are not compared with, other than table 1. Why is there no GAN in table 1, given that your section 3 seems to imply that GAN is the most related? Also, the quality of presentation of the figures is not great.

Confidence in this Review

2-Confident (read it all; understood it all reasonably well)


Reviewer 3

Summary

Authors present a method similar to teacher forcing that uses generative adversarial networks to guide training on sequential tasks.

Qualitative Assessment

Cute name. Work is a creative idea to improve upon the concept of teacher forcing. Experimental work focusses on three fairly diverse problems. Results are convincing to me.

Confidence in this Review

2-Confident (read it all; understood it all reasonably well)


Reviewer 4

Summary

This paper proposes a method for training recurrent neural networks (RNN) in the framework of adversarial training. Since RNNs can be used to generate sequential data, the goal is to optimize the network parameters in such a way that the generated samples are hard to distinguish from real data. This is particularly interesting for RNNs as the classical training criterion only involves the prediction of the next symbol in the sequence. Given a sequence of symbols x_1, ..., x_t, the model is trained so as to output y_t as close to x_{t+1} as possible. Training that way does not provide models that are robust during generation, as a mistake at time t potentially makes the prediction at time t+k totally unreliable. This idea is somewhat similar to the idea of computing a sentence-wide loss in the context of encode-decoder translation models. The loss can only be computed after a complete sequence has been generated.

Qualitative Assessment

The idea in this paper is interesting and well motivated. When training an RNN for generation, using the likelihood of the observed data is not the proper criterion. Even though the problem and approach are interesting, I find the description of the training objective (Sec. 2) very confusing and unclear. Precise remarks follow: - The random variable y is present in two expectations in Eq. (1). - The function B takes y as an input, but actually does not need to. Given an RNN with a given sequence of inputs, all the y can be computed. Moreover, taking the expectation with respect to y following the distribution parametrized by \theta_g does not make sense. Also, I guess that the authors use an empirical approximation of this expectation. It might be cleared to write it as an average on training samples. - The adversarial training is a min-max problem (Googfellow et al., 2014). The authors state that they "perform SG steps on ..." and "always do gradient steps on ...". This means that the two parameters are optimized using different objective functions. One could probably get the same results by maximizing -log(D) instead of minimizing -log(1-D). This would allow the authors to write the optimization problem they try to solve. - The different terms of the objective are not weighted in any way. It would be interesting to see how a trade-off between NLL and the other terms plays in terms of final performance. If the adversarial objective is seen as a "regularizer" (l. 56), it would be good to use a weighting term. The experimental evaluation regarding the distribution of hidden states and the t-SNE maps is a bit frustrating. No discussion in the paper is proposed and the only explanations are in the caption of Fig. 3. From what I understood, it simply shows the fact that hidden states diverge when generating sequences, as the input shifts away from the true data distribution. More discussion would be welcome. As far as the experimental evaluation is concerned, the experiments could be more convincing. For the character-level language modeling, the paper compares models trained with and without the adversarial loss. An improvement potentially due to the regularizing effect is shown (0.02 Bits/character). My concern is that the baseline model is not trained properly. Other papers report much lower performances on Penn Treebank, see for instance a recent workshop paper at ICLR2016: Alternative Structures for Character-Level RNNs, Piotr Bojanowski, Armand Joulin and Tomas Mikolov (Table 1). The model they used was a plain RNN and it achieved a validation BPC of 1.42. When the baseline model is not properly tuned, it is hard to assess the impact of the regularization. Overall this is an paper tackling an interesting and important problem. However, the presentation is a bit overly-complicated and unclear. Experiments exhibiting improved likelihood (imputed to regularization) on held-out data are not convincing and the third experimental claim (tSNE maps) is not properly discussed in the paper. Of course, we need to keep in mind that properly evaluating the performance of generative models is complicated, and the user study shows statistically significant improvements.

Confidence in this Review

2-Confident (read it all; understood it all reasonably well)


Reviewer 5

Summary

The paper proposes a novel approach to train RNN using adversarial network to enforce its self-generated sequence to be similar to that of the ground-truth training sequence. Traditional approach to train RNN assumed that when predicting output at time t, ground truth output sequence of time 1 to t-1 is given. However, this is not guaranteed at test time since the output of time 1 to t-1 is self-generated. The paper tries to solve this issue by enforcing the behavior of self-generating RNN to be similar to that of ground-truth RNN using adversarial network. This approach is especially helpful when generating longer sequence than what the network has seen in the training data.

Qualitative Assessment

1. I am excited to find such a simple concept improving RNN. I have experience with RNN suffering from generating bad results after the length of sequence it has seen in training phase, and this paper clearly suggests a solution for this issue. 2. Experiments seem to be overall well-designed in various perspective, although it was not satisfactory to not see quantitative result significantly outperforming baselines. However, qualitative result of section 4.4 clearly shows the contribution of this work. It would have been better if there was an experiment quantitatively showing the result similar to 4.4, especially since this part seems to be the important contribution of this work. 3. In practical stance, I am worried that this idea may not be well-appreciated, especially since machine learning tasks tend to be trained and evaluated on the same dataset, which has fixed distribution of the output sequence. Although the experiments shows that this work helps improving such test results, the gain seems marginal while implementing adversarial network for it seems like a overkill. It may be more practical to implement domain-specific idea and increase the model size like what Pixel RNN did to outperform this work. 4. I am having difficulty finding the loss definition in your experiments. You seem to have used $NLL+C_f$ or $NLL+C_f+C_t$ according to definition in 2.2. However, it is not obvious to me which loss you used for the experiments and which performs better in what kind of cases when I want to implement one myself. Am I missing something?

Confidence in this Review

2-Confident (read it all; understood it all reasonably well)


Reviewer 6

Summary

The authors present a method to stabilize sequence generation models by introducing an adversarial network which forces the distribution of hidden state trajectories during generation to match that during training. They evaluate this on several datasets, including character-level language modeling, sequential pixel generation for MNIST, handwriting synthesis and synthesis of waveforms. They provide a visualization which shows that the distributions do indeed match better with the additional adversarial loss than without.

Qualitative Assessment

The paper addresses an important problem seen during sequence generation, which is that recurrent models are unstable and their hidden states often drift which makes their generations incoherent over long timescales (even though they may be locally coherent). The method they introduce is very straightforward but seems to work decently well. This is evidenced by improvements in negative log-likelihood, improved quality of generated samples (according to human annotators) and a visualization experiment which shows that the hidden states during generation occupy a similar region as the hidden states during training, unlike RNNs trained the usual way. The paper is overall very well written and clear. Comments: -There is no evaluation of the handwriting generation experiments beyond showing some generated samples - at least have human annotators evaluate them as for the vocal synthesis experiments. Are the shown samples random and not cherry-picked? Figure 6: specify this is for music synthesis experiments.

Confidence in this Review

2-Confident (read it all; understood it all reasonably well)